# Highly Sensitive THz Gas-Sensor Based on the Guided Bloch Surface Wave Resonance in Polymeric Photonic Crystals

**DOI:** 10.3390/ma13051217

**Published:** 2020-03-08

**Authors:** Chi Zhang, Suling Shen, Qiong Wang, Mi Lin, Zhengbiao Ouyang, Qiang Liu

**Affiliations:** 1THz Technical Research Center, Shenzhen University, Shenzhen 518060, China; chichi1995@126.com (C.Z.); qwang@szu.edu.cn (Q.W.); linfengas111@szu.edu.cn (M.L.); zbouyang@126.com (Z.O.); 2College of Physics & Optoelectronic Engineering, Shenzhen University, Shenzhen 518060, China; 3National-Regional Key Technology Engineering Laboratory for Medical Ultrasound, Guangdong Key Laboratory for Biomedical Measurements and Ultrasound Imaging, School of Biomedical Engineering, Health Science Center, Shenzhen University, Shenzhen 518060, China

**Keywords:** bloch surface wave, gas sensor, polyvinylidene fluoride (PVDF), polycarbonate (PC), grating, photonic crystals

## Abstract

THz waves have interesting applications in refractive index sensing. A THz gas sensor based on the guided Bloch surface wave resonance (GBSWR) in a one-dimensional photonic crystal (1DPhC), which consists of periodic polycarbonate (PC) layers and polyvinylidene fluoride (PVDF) layers, has been proposed. Numerical results based on finite element method (FEM) show that the photonic band gap that confines Bloch surface waves (BSWs) lies in the regime of 11.54 to 21.43 THz, in which THz wave can transmit in both PC and PVDF with the ignored absorption. The calculated sensitivity of hazardous gas HCN in angle is found to be 118.6°/RIU (and the corresponding figure of merit (FOM) is 227) and the sensitivity in frequency is 4.7 THz/RIU (the corresponding FOM is 301.3). The proposed structure may also be used for monitoring hazardous gases which show absorption to the incident THz wave. Further results show that for N_2_O gas, the maximum sensitivity goes up to 644 (transmittance unit/ one unit of the imaginary part of the refractive index). The proposed design may find applications in the detection of dangerous gases.

## 1. Introduction

Gas sensors have wide applications in toxic monitoring, explosive detection and pollution control. One method for detecting dangerous gas is based on surface plasmon resonance (SPR) [1,2,3]. When the surface plasmonic polaritons (SPPs) along the metal/dielectric interface are excited with wave vector that matches SPR, sensitive gas sensing can be achieved. On the other hand, due to properties such as low-loss and availability for scaling up and down, bound states in the continuum (BICs) or “dark optical states” sustained within the all-dielectric micro/nanostructures can provide another platform for sensing applications, in which surface polaritons that propagate at the interface of dielectric gratings can be utilized for trapping and manipulating photons [4,5,6].

THz wave, covering the electromagnetic spectrum from 1 to 10 THz, has wide applications in prospects of studying various chemical gases since the rotational or transitional energy of most gaseous molecules lies in such range [7]. Flammable gases such as N_2_O, HCN, H_2_ have strong absorption peaks in terahertz bands which are also termed as “fingerprints.” For example, according to rotation-vibration spectra, the absorption peak of hazardous gas molecular HCN lies at 1.24 THz [8].

Gas sensors based on graphene have been extensively studied since SPPs can be supported in grapheme in the range of THz wave. Compared with noble metals, graphene shows distinguished properties such as less intrinsic loss and the flexible tunability that realized by the chemical and electrical doping methods. Due to the relative large refractive indices of graphene in the THz regime, the figure of merit (FOM) in THz gas sensor is typically large (e.g., above 1150 /RIU in Table 1); however, the sensitivity in angle (the angle shift as in response to the change of refractive index) is typically small (e.g., 34.11°/RIU in Table 1). In addition, since the imaginary part of permittivity of graphene is large in the THz range, prism with very high refractive index or an air gap in between the graphene and prism is usually required [9], which makes the whole sensing system bulky and may hinder the potential application. In this paper, by introducing all polymeric gratings instead of prism on the 1D-PhC, higher FOM and high sensitivity can be numerically achieved at the same time.

Bloch surface wave (BSW) could be classified as one kind of BICs, it has applications in refractive index sensing area [10,11,12]. Quite similar to SPPs [13], BSWs propagating along the interface between the dielectric layer and the one-dimensional photonic crystals (1DPhCs), strong enhancement in electric field can be produced at the interface due to the bandgap effect. Since there are a variety of dielectric materials that can be selected to construct 1DPhCs, the dispersion of BSW can be arbitrarily tuned, as compared with the case in SPP that provided by limited novel metals [14]. In addition, because of the low loss property of dielectric materials, relatively narrower resonant peaks can be obtained in BSW-based structures.

In this paper, by combing with the merits of THz wave and BSW, a new kind of THz gas sensor is designed with polymeric materials that are inexpensive to fabricate. According to the simulation results based on finite element method (FEM), the proposed gas sensor is sensitive to the change of refractive index real part of HCN, as indicated by the high FOM measured by frequency ~147.3 /RIU and FOM measured by the incident angle ~301.3°/RIU. The dispersion of BSW has been discussed to explain the sensing mechanism. The above mentioned sensing mechanism is based on the change of the real part of the refractive index, in which the analyte does not show absorption to the incident THz wave. In the following, we present discussions on another way of gas sensing which is sensitive to the change of the imaginary part of the refractive index, in such cases the analyte shows absorption to THz wave. Such as N_2_O, for gases with an absorption spectrum in the 10–20 THz band, gas sensing applications can be realized. At the end of this paper, comparisons in terms of sensitivity, fabrication materials, operation frequency and gas contents have been conducted and the potential applications of the proposed design have been discussed.

## 2. Structure and Analysis

The schematic of BSW-based THz gas sensor is shown in Figure 1. The proposed dielectric multilayer sustains transverse electric (TE)-polarized (transverse magnetic (TM)-polarized light can also be supported) BSWs which has N = 9 alternating dielectric layers of polyvinylidene fluoride (PVDF) (denoted with light green color, with a thickness of *t*_PVDF_ = 7 μm) and polycarbonate (PC) (denoted with gray color, with a thickness of *t*_PC_ = 5 μm). The buffering layer (with the thickness of *t*_buf_ = 24 μm), which is composed of a high refractive index PC material, is introduced at the bottom of the PhC structure and a coupling grating is introduced under the buffering layer (with grating constant Λ = 10 μm, height of grating *h*_g_ = 10 μm and aspect ratio AR = 0.28). Both the buffering layer and the grating are made of PC material. The sensor is surrounded by gas, with refractive index *n_g_.* During operation, when the sensor is placed in the atmosphere of the detected gas, a terahertz wave is incident from the bottom in a direction at an angle θ to the normal. Because the change in the refractive index of the gas produces different degrees of perturbation, the resonance frequency shifts and the parameters of the excited BSW are also changed. The electromagnetic waves are exponentially elapsed in the vertical direction in the photonic crystal layer.

In the terahertz region, the PVDF dielectric function exhibits a resonance,
(1)$$\varepsilon_{PVDF}(\omega )=\varepsilon_{opt}+\frac{(\varepsilon_{dc}-\varepsilon_{opt})\omega_{TO}^{2}}{\omega_{TO}^{2}-\omega^{2}+i\gamma \omega }$$
where ε_opt_ = 2.0, ε_dc_ = 50.0, ω_TO_ = 0.3 THz and γ = 0.1 THz [15]. In comparison with PVDF, the dielectric response of PC polymer is frequency independent, having a purely real dielectric constant ε_PC_ = 2.56. The high and low refractive index layers are PC and PVDF materials, respectively and their dielectric constants are 2.56 and 1.9570-0.0004*i* at 10 THz.

When material losses are negligible and the number of reflector periods is infinite, it indicates that there exists a frequency range for complete TE polarization reflection for a given angle of incidence, according to the theory of planar periodic reflector SPR. The formula of the wave vector is given as follows:(2)$$k_{BSW}=-k_{0}n_{0}\sin(\theta )+\frac{2\pi m}{\Lambda }$$
where *k*_0_ = *2*π*/*λ and *k*_BSW_ are the magnitudes of the free space wave vector and grating BSW wave vector respectively, *n*_0_ is the refractive index of the sensing gas, θ is the incident angle, Λ is the grating period and m is an integer [16,17], the negative sign “−” means the propagation direction of the BSW is opposite that of incident *k*_0_ vector that horizontally projected. As THz wave is incident on the grating, it produces the reflected and BSW waves. As according to References [16,17,18], the direction of the BSW wave is opposite to the direction of the x component of the incident wave, it thus originates the negative sign as shown in Equation (2). In this paper, *m* = 1 and the incident angle is 0° < θ < 90°. The wavelength range corresponding to a certain grating constant can be obtained as follow:(3)$$\frac{k_{BSW}}{k_{0}}<\frac{\lambda }{\Lambda }<\frac{k_{BSW}}{k_{0}}+1$$
when Λ = 10 μm, the effective index of refraction 1.4 < *n*_eff_ < 1.6, from Equation (3). Therefore, the wave numbers of BSW is in the range of 1.4*k*_0_ to 1.6*k*_0_. We can get the operation frequency *f* for producing BSW, which lies at the range: 11.54 THz < *f* < 21.43 THz, such analytical result is close to the simulation result as indicated by the frequency range of the purple line that shown in Figure 2a.

Figure 2a,b shows the band diagrams of the periodic PVDF/PC dielectric structure for TE and TM polarized modes. The numerical simulations were performed by means of the finite element method [19]. The white regions represent the 1D PhC forbidden band. By introducing the buffer layer that destroys the periodicity of the 1D PhC, it is possible to excite the BSW which only appears within the band gap (see the pink line). Compared with TE mode, TM mode has a much narrower band gap due to the Brewster effect; in addition, the frequency range for observing BSW in the same structure is quite limited (see the blue line in Figure 2b). Figure 2c shows that for the TE case, when the refractive index of the background *n*_0_ changes, the *k_x_* of the excited BSW is varied. According to Equations (2) and (3), the *k_x_* is directly connected with the incident angle and the operation frequency (please refer it to the dispersion diagram shown in Figure 2a), by measuring the incident angle or operation frequency, the aim of sensing *n*_0_ may be achieved.

When *k_x_* and frequency satisfy certain condition (please refer it to the pink line shown in Figure 2a), it is possible to produce BSW. The electric field distribution of the BSW mode in the *y* direction is shown in Figure 3. It can be seen that the electric field is mainly concentrated in the buffer layer and it gradually decreases along +*y* directions. The electric field distribution of TE mode should satisfy the following formula:
(4)$$E(x,y,z)=\sum_{m=-\infty }^{\infty }\sum_{n=-\infty }^{\infty }S_{m,n}(y)e^{-j(k_{x,m}x+k_{z,n}y)}$$
where,
(5)$$k_{x,m}=k_{x,inc}-\frac{2\pi m}{\Lambda_{x}},m=-\infty ,\ldots ,-2,-1,0,1,2,\ldots ,\infty ,$$
(6)$$k_{z,n}=k_{z,inc}-\frac{2\pi n}{\Lambda_{z}},n=-\infty ,\ldots ,-2,-1,0,1,2,\ldots ,\infty ,$$
(7)$$k_{x,inc}=\frac{2\pi }{\lambda }n_{0}\sin(\theta_{inc})\cos(\phi ),k_{z,inc}=\frac{2\pi }{\lambda }n_{0}\sin(\theta_{inc})\sin(\phi )$$
*k_x,inc_* and *k_z,inc_* are the *x* and *z* components of *k_inc_* Λ*_x_* and Λ*_z_* are the grating period in the corresponding direction. *S*_m,n_(*y*) is the Fourier coefficients, which can be computed by solving Maxwell’s equations in Fourier space [20].

By tuning the concentration and (or) pressure of the measured gas, the dielectric constant can be varied. Such phenomenon may be described by the equation as given as [8]:(8)$$\varepsilon =\varepsilon_{\infty }+\sum_{i=1}^{\infty }\frac{\Delta \varepsilon_{i}(C,P)\omega_{0i}^{2}}{\omega_{0i}^{2}-\omega^{2}-i\gamma (P)\omega }$$
where we select the HCN as the measured gas, note that $\varepsilon_{\infty }=1$ is the background dielectric constant. $\omega_{0i}$ is the angular frequency corresponding to the absorption peak. $\gamma_{i}(P)=P\times 2\times \pi \times 6$ MHz/hPa is the damping rate. $\Delta \varepsilon_{i}(C,P)$ is the absorption strength factor, which is a function of pressure *P* and concentration *C*. Given that the frequency of the incident THz wave is 14THz, the refractive index of HCN is: *n*_0_ = 1.000 + 2.14 × 10^−7^i. By increasing pressure or concentration, the refractive index of the gas can be tuned; here we assume the real part of the *n*_0_ changes from 1.000 to 1.003, apparent shifts in the spectra can be found in Figure 4a,b, where *S*_θ_ decreases with the increase of *n*_0_, that is, *S*_θ_ and FOM can reach 123°/RIU and 147.3 /RIU, respectively, which can come up with following formula:(9)$$S_{\theta }=\frac{\Delta \theta }{\Delta n},S_{f}=\frac{\Delta f}{\Delta n}$$
(10)$${\mathrm{FOM}}_{\theta }=\frac{S_{\theta }}{\theta_{FWHM}},FOM_{f}=\frac{S_{f}}{f_{FWHM}}$$

The change is because the angle sensitivity is proportional to the reciprocal of the incident cosine value. FOM has the same but very slow change of *S*_θ_, which is shown in Figure 4c. It can be seen that as *n*_0_ increases by 0.01, FOM only decreases 4.1/RIU. Similarly, the phenomena can also be observed in Figure 4b,d, when wave is incident at a fixed 60°. By observing the shift of frequency spectrum, the change of *S*_f_ is not obvious in Figure 4b and FOM increases with the increase of *n*_0_, that is, *S*_f_ and FOM can reach 4.7 THz/RIU and 301.3 /RIU, respectively. This sensor can simultaneously reach a high and stable sensitivity and FOM in both angular and frequency domain. It needs to be pointed out that, since the contents of the top layer and substrate are PC, it is easy to fabricate micro-porous structure at the interface of buffer layer [21]. By doing so, more polar molecules may be adsorbed to the relatively large surface, which may enhance the sensitivity [22]; similarly, as more un-polar molecules are filled in the porous space of PC surfaces, the effective index of refraction of the top layer can also be adjusted, which in turn help enhance the sensitivity.

At present, most researches on THz gas sensor focus on sensing the change of real part of the analyte, see References [1,2,3,7]. The structure based on BSW as proposed here may also be applied for detecting the minor change of the imaginary part of the refractive index of gas. In the area of atmospheric pollution and dangerous gas detection, gases such as N_2_O, HCN, H_2_ and CO_2_ have obvious absorption peaks in the THz band. From Equation (8), it can be seen that, when the operation frequency of the sensor is consistent with the absorption frequency of gas molecules, in which ω = ω_0*i*_, different concentration *C* and (or) pressure *P* will make the imaginary part of the refractive index of analyte be varied; by measuring the change of transmission in the spectra, the sensing to the imaginary part may also be achieved. Here we select the gas N_2_O with the absorption peak appeared in 17.664 THz (which is equivalent to the 588.726 cm^−1^) to detect the change of imaginary part of the N_2_O gas [23]. Since there is no reference that can show the relationship between $\Delta \varepsilon_{i}(C,P)$ and concentration C (or pressure P), in the following we consider four cases where the imaginary part is 0, 0.00002, 0.0002 and 0.002 respectively. The transmission spectrum as simulated with finite element method is shown in Figure 5. It can be found that the transmittance is sensitive to the minor change of the imaginary part of the refractive index (e.g., the imaginary part changes from 0 to 0.002, the transmission changes from nearly 0 to 0.15) and the maximum sensitivity can reach 664 (transmittance unit/ one unit of the imaginary part of the refractive index). Further optimization in terms of increasing the number of photonic crystal layers, replacing PVDF with another polymer film that shows higher refractive index may also increase the sensitivity. It turns out that our design is not only sensitive for sensing the real part of the refractive index of the gas but also be applied as monitoring the minor change of imaginary part of the refractive index of the analyte. Consider that hazardous gases such as N_2_O, HCN, H_2_ and CO_2_ show “fingerprints” in the THz band, the proposed design may find certain applications in sensing the concentration or pressure of such gases.

We discussed the real-part-refractive-index sensing of HCN at 14 THz and the imaginary-part-refractive-index sensing of N_2_O near the absorption spectrum. This proves the applicability of the proposed structure. In recent theoretical terahertz sensing articles, only a few have explored sensing applications for practical gases. Performance comparison between the proposed BSW based design and other reported THz gas sensors is summarized in Table 1. References [1,2,3] all discuss the sensing sensitivity of different refractive indices and only Reference [7] discusses the concentration sensing of H_2_. It is difficult to find the relationship between the corresponding dielectric constant and concentration (pressure) of the corresponding gas, so it is hard to do practical inquiries into specific gases. For the comparison of sensing performance, angular sensitivity and the value of *S*_f_/*f* can be applied. The *S*_θ_ has a good performance compared to References [1,2,3]. However compared to Reference [7] in terms of frequency sensitivity, the value of *S*_f_/*f* can only reach 0.34 /RIU. It is noted that since the refractive index contrast between PC and PVDF is quite limited ~1, which is much smaller than the case in silicon and air (3.4:1) as reported in Reference [7], the normalized sensitivity of frequency *S*_f_/*f* in the proposed design is small. However, the sensitivity in angle as shown in the BSW-based THz sensor is relatively high; in addition, considering the availability for sensing the minor change of both real and imaginary parts of refractive index of gases, our design may find potential applications in sensing hazardous gases in the terahertz band.

## 3. Conclusions

BSW has the advantage of low loss compared with SPR. However, there is not much research in THz range. Using polymer materials can effectively reduce the loss of THz wave and also cut the fabrication costs. In this paper, a terahertz gas sensor by using polymer materials of PVDF and PC based on BSW is proposed. The photonic band structure of BSW was studied and the discussion on the working mechanism is provided with details. The BSW excitation frequency range corresponding to different grating constant was calculated. The BSW structure we show has an angular and frequency sensitivity of 123°/RIU and 4.7 THz/RIU with FOM of 147.3 /RIU and 301.3/RIU, respectively. In addition to sensing the change of the real part of the refractive index, the proposed all-polymer gas sensor is also sensitive to the change of the imaginary part of the refractive index. We take the N_2_O gas for an example, the maximum sensitivity goes up to 644 (transmittance unit/ one unit of the imaginary part of the refractive index). The proposed gas sensor has main advantages as high sensitivity and cost-effective. Such design may be used in the area of non-contact sensing of dangerous and hazardous gases.

## Figures and Tables

**Figure 1 materials-13-01217-f001:**
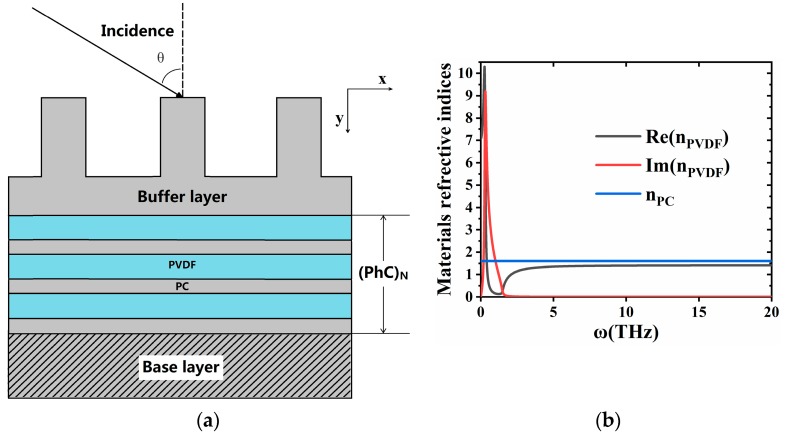
(**a**) Schematic illustration of the dielectric multilayer which consists of 9 alternating dielectric layers of polycarbonate (PC) (7 μm) and; polyvinylidene fluoride (PVDF) (5 μm). (**b**) Dielectric constant of PVDF and PC materials in terahertz band (the black and red solid line represents real and imaginary parts of dielectric constant of PVDF material; the blue line shows the PC material).

**Figure 2 materials-13-01217-f002:**
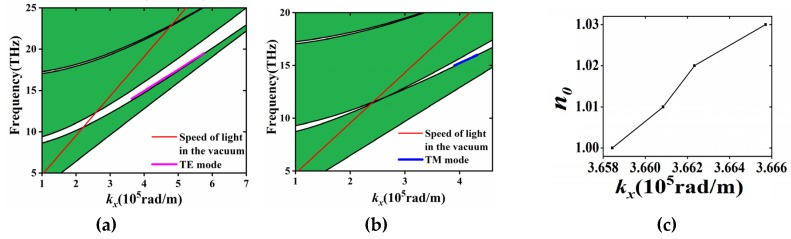
The dispersion relation of the dielectric multilayer for (**a**) TE and (**b**) TM polarization (the white zone denotes the stop band and the blue zone denote the non-radiative modes for the BSW and the red solid lines represents the speed of light in a vacuum, while the green point shows the working mode) when *t*_buf_ = 24 μm, *t*_bas_ = 80 μm. (**c**) *n*_0_ versus *k_x_* when the frequency of the incident terahertz wave is 14 THz, provided with the TE polarization.

**Figure 3 materials-13-01217-f003:**
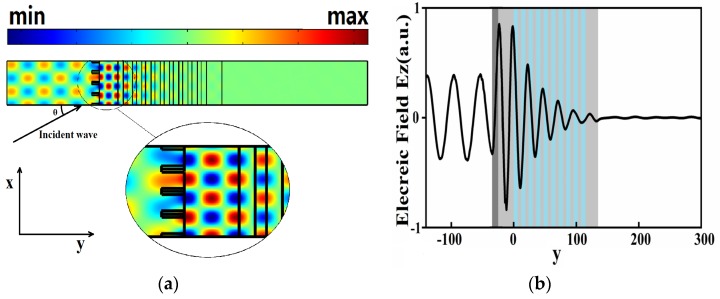
(**a**)Electric field distribution in the y direction in the sensor Bloch surface wave (BSW) mode with the incident angle of 60°. (**b**) The electric field component in the z-direction along the *y*-direction, which is the top edge of the structure in (**a**) (the dark gray part is the position of the grating).

**Figure 4 materials-13-01217-f004:**
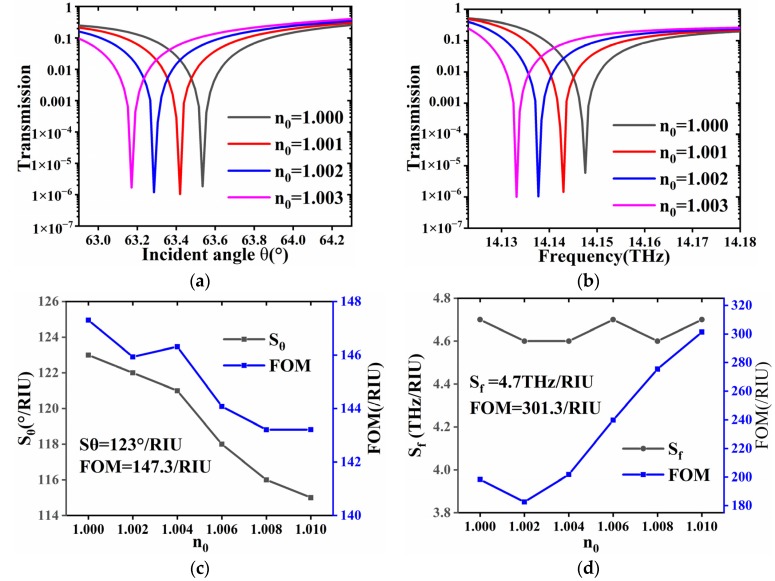
Performance of sensing effect in frequency and angle. (**a**,**b**) show the variation of transmittance in angular and frequency spectrum under different *n*_0_. (**c**,**d**) extract the sensitivity and FOM of the Transmission spectrum corresponding to different refractive index gases from 1.00 to 1.01.

**Figure 5 materials-13-01217-f005:**
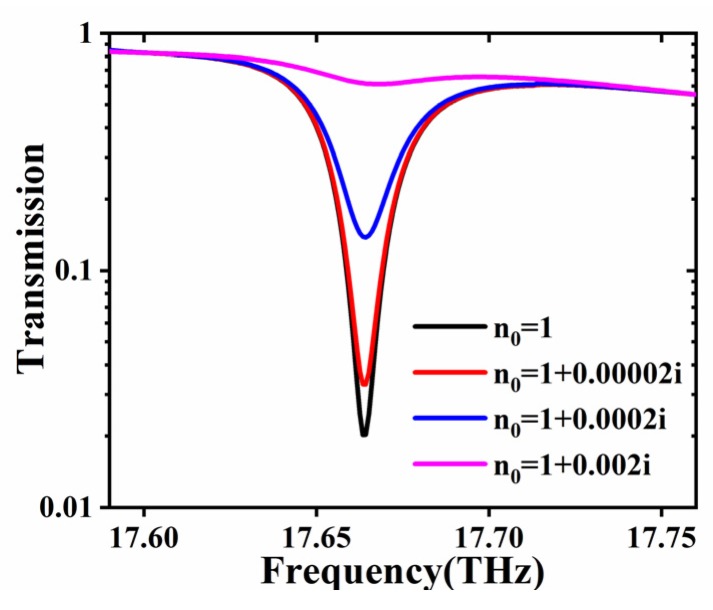
The transmission spectrum of the proposed structure when the background gas is N_2_O Here the imaginary parts of refractive index is changed from 0 to 0.002. Following the scaling law of electromagnetic theory, the geometry of the structure shown in Figure 1 has been scaling down to nearly 80%, noted that the operation frequency is 17.664 THz corresponding to the absorption peak of N_2_O molecule, here the incident angle is assumed to be 60°.

**Table 1 materials-13-01217-t001:** Comparison of the recently reported THz gas sensors.

Sensor	*S*_f_ (Hz/RIU)& FOM(RIU^−1^)	*S*_θ_ (°/RIU)& FOM(RIU^−1^)	Material	Working Frequency *f*	Analyte
Doped graphene THz plasmonic gas sensor [1]	N/A	*S*_θ_ = 34.11°/RIU FOM = 1150 /RIU	Doped Graphene	1~5 THz	N/A
Graphene SPR gas sensor [2]	N/A	*S*_θ_ = 40.6°/RIU FOM = 741/RIU	Graphene	1~5 THz	N/A
Terahertz gas sensor based on SPR with graphene [3]	N/A	*S*_θ_ = 147°/RIU FOM = 10 /RIU	Graphene	2.5~5 THz	N/A
One-dimensional photonic crystal terahertz gas sensor [7]	*S*_f_ = 0.3 THz/RIU FOM = 6000 /RIU *S*_f_/*f* = 1 /RIU	N/A	Silicon	290~360 THz	H_2_
This work	*S*_f_ = 4.7 THz/RIU FOM = 147.3 /RIU *S*_f_/*f* = 0.34 /RIU	*S*_θ_ = 123°/RIU FOM = 301.3/RIU	Polymer (PVDF, PC)	10~20 THz	HCN

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
