# Peer review of "Highly Sensitive THz Gas-Sensor Based on the Guided Bloch Surface Wave Resonance in Polymeric Photonic Crystals"

_materials, 2020, doi:10.3390/ma13051217_

Round 1
Reviewer 1 Report
The Authors present the numerical investigation of a sensor based of the excitation of surface states in a dielectric photonic crystal structure. Although the idea is not very novel, the results are interesting and the manuscript reads well. The discussion is a little bit short and needs be expanded. My comments are:
Line 3. There is no need for a paragraph break there. In the introduction the Authors should comment more of the surface polaritons sustained in dielectric media/gratings as they can provide many interesting sensing and wavefront manipulation functions. Some additional material/references are:
-Bound states in the continuum, Nature Reviews Materials 1 (9), 16048
-Frequency splitter based on the directional emission from surface modes in dielectric photonic crystal structures, Optics express 23 (11), 13972-13982
-Nonradiating photonics with resonant dielectric nanostructures, Nanophotonics 8 (5), 725-745
-Bound states in the continuum, Nature Reviews Materials 1 (9), 16048
The equation at line 112 should be better integrated in the in line text Figure 3b. At which point along the x axis is the electric field calculated. Also in Figure a the field does not appear to penetrate the crystal but in Figure 3b there seems to be some significant field there. Maybe the Authors could expand the axis of Figure 3b to show the evanescent nature of the surface waves. How many unit cell are depicted in Figure 3a. Maybe detail image around the grating would help Towards expanding the discussion the Authors could include also the TM polarization
Author Response
Response to Reviewer 1 Comments
The new version of the article is shown in the attachment.
Please see the attachment.
Point 1: Line 3. There is no need for a paragraph break there.There is no need for a paragraph break there. In the introduction the Authors should comment more of the surface polaritons sustained in dielectric media/gratings as they can provide many interesting sensing and wavefront manipulation functions.
-Bound states in the continuum, Nature Reviews Materials 1 (9), 16048
-Frequency splitter based on the directional emission from surface modes in dielectric photonic crystal structures, Optics express 23 (11), 13972-13982
-Nonradiating photonics with resonant dielectric nanostructures, Nanophotonics 8 (5), 725-745
Response 1: That is the name of Bloch surface wave, not a paragraph break.
We have added the comment as below. Line 33: “On the other hand, due to the properties as low-loss and availability for scaling up and down, bound states in the continuum (BICs), or “dark optical states” that sustained within the all-dielectric micro/nano structures can provide another platform for sensing applications, in which surface polaritons that propogating at the interface of dielectric gratings can be utilized for trapping and manipulating photons”.
Point 2:The equation at line 112 should be better integrated in the in line text Figure 3b. At which point along the x axis is the electric field calculated. Also in Figure a the field does not appear to penetrate the crystal but in Figure 3b there seems to be some significant field there. Maybe the Authors could expand the axis of Figure 3b to show the evanescent nature of the surface waves.
Response 2:We added the formula Eq.(4)-(7) to make it easier to read.
Fig. 3b is an electric field intensity diagram along the upper surface of Fig. 3a, we have added "The electric field component in the z-direction along the y-direction, which is the top edge of the structure in (a)." as shown in the contents starting from the line 129.
We expanded the calculation range and optimized the thickness of the base layer to make the evanescent nature of the surface waves more visible. It’s shown in Fig.3a.
Point 3: How many unit cell are depicted in Figure 3a. Maybe detail image around the grating would help Towards expanding the discussion.
Response 3: In order to prevent a single cycle from being insufficient to show its characteristics, we set up a structure of 4 cycles to draw Fig.3a.
We have magnified the grating part. Due to the incident angle, the grating and BSW are not well coupled, and the generated electric field will be partially distorted. This will be our next step work.
Point 4: the Authors could include also the TM polarization.
Response 4: In the band diagram of Fig.2, we have added the band diagram of TM mode Fig.2 (b). We revised as”By introducing the buffer layer that destroys the periodicity of the 1D PhC, it is possible to excite the BSW which only appears within the band gap (see the pink line). Compared with TE mode, TM mode has a much narrower band gap due to the Brewster effect; in addition, the frequency range for observing BSW in the same structure is quite limited (see the blue line in Fig. 2(b).” as shown from line 114 to 118.

Reviewer 2 Report
Bloch surface waves starting widely applied in the field of bio- and gas sensing technologies from optical to THz wavelengths range. Thus, this is interesting and actual topic nowadays.
However, I would like to propose the authors to consider a few critical comments below:
1.PVDF and PC abbreviations are not mentioned in the text.
2.Would be useful to explain why minus sign appear in the expression (2)?
3.“Figure 2. The projected band structure of the dielectric multilayer for TE polarization (The white zone denotes the stop band and the blue zone denote the conduction band for the BSW and the red solid lines represents the speed of air, while the green point shows the working mode.). “
I would say that the authors uses not correct concepts in term of this topic: projected band structure – dispersion relation, conduction band – authors do not have any metals or absorbing materials in the working region! For the surface waves it’s called non-radiative modes. And, finally, speed of air – I would guess that, this is speed of light in the air or vacuum.
4.I would suggest to redraw the graph and clearly indicate the edges of the band gap of photonic crystal and the dispersion line of the BSW inside.
5.Fig.2b mentioned in the text, however, it is not highlighted in the Figure 2.
6.Fig. 3a should be more clearly presented, is too small.
7.What kind of gases the authors tested in their study? The authors presented only simulation where refractive index of air is changed. Please, use the real refractive index dispersions of gases in your simulations. Moreover, in the introduction authors mentioned that transitional and rotational bands of molecules in the gases have absorption in the THz region. For what kind of specific gases in the 10-20 THz region such sensor would be applied?
8.The authors compared their simulated sensors response with presented in literature, however, the range of frequencies is rather different, what leads to detection of distinct type of gases, and sensitivity comparison become difficult. So I would like to suggest the authors to find one type of gases, which they can simulate and compared with literature.
9.Would be very useful to analyse the cases with polar and non-polar gases, some references could be find here: A. Nooke et. al. Sensors and Actuators B 149 (2010) 194–198.
And a freshly published comparison of sensitivity features between BSW and SPR should be also cited:
Z. Balevicius and A. Baskys, Optical Dispersions of Bloch Surface Waves and Surface Plasmon Polaritons:Towards Advanced Biosensors, Materials 2019, 12, 3147; doi:10.3390/ma12193147
Please, check grammar quite much mistakes with missing of letters.
The manuscript is rather short and I would like to suggest the authors to expand the discussion part in the light of critical comments presented above. The manuscript cannot be published in such form.

Author Response
Response to Reviewer 2 Comments
The new version of the article is shown in the attachment.
Please see the attachment.
Point 1: PVDF and PC abbreviations are not mentioned in the text. 

Response 1: The full name is only mentioned in the abstract. There is indeed a case where the full name is not mentioned in the text. It has been added on lines 79 and 80.
Point 2: Would be useful to explain why minus sign appear in the expression (2)?
Response 2: The negative sign before k0 in formula (2) is because the incident wave is irradiated on the grating, which generates reflection and forms BSW on the surface. Therefore, the generated BSW is opposite to the x-direction component of the incident wave. It can be seen in the reference that the direction of the generated surface wave is opposite to the direction of the x component of the incident wave. Some literatures do not add a negative sign because the kBSW is negative. In this article, the negative sign is extracted, and kBSW is positive.
Maier, Stefan A. Plasmonics: Fundamentals and Applications. Springer US, 2007.
Daniel P. C., Andrew R. N.. Engineering surface plasmon grating couplers through computer simulation. Journal of Vacuum Science & Technology B 2008, 26, 2183-2187.
Stephan T. K., Amit A., Henri J.L. Vladimir A. A.. An Efficient Large-Area Grating Coupler for Surface Plasmon Polaritons. Plasmonics 2012, 7,269-277.
Point 3: “Figure 2. The projected band structure of the dielectric multilayer for TE polarization (The white zone denotes the stop band and the blue zone denote the conduction band for the BSW and the red solid lines represents the speed of air, while the green point shows the working mode.). ” I would say the authors uses not correct concepts in term of this topic: projected band structure – dispersion relation, conduction band – authors do not have any metals or absorbing materials in the working region! For the surface waves it’s called non-radiative modes. And, finally, speed of air – I would guess that, this is speed of light in the air or vacuum.
Response 3: Strongly agree with your proposal, and have modified the corresponding concepts in the article.
We revised as "The dispersion relation of the dielectric multilayer for (a) TE and (b) TM polarization (The white zone denotes the stop band and the blue zone denote the non-radiative modes for the BSW and the red solid lines represents the speed of light in the vacuum, while the green point shows the working mode.) when tbuf = 24 μm, tbas = 80μm." as shown in the contents starting from the line 123 to line 126.
Point 4: I would suggest to redraw the graph and clearly indicate the edges of the band gap of photonic crystal and the dispersion line of the BSW inside.
Response 4: A clear dispersion relation has been drawn at Fig.2(a), and the TM mode has been added at Fig2.(b).
Point 5: Fig.2b mentioned in the text, however, it is not highlighted in the Figure 2.
Response 5: Fig. 2b refers to the kBSW generated by gases with different refractive indices in the TE mode and has been modified to Fig. 2c.
Point 6: Fig. 3a should be more clearly presented, is too small.
Response 6: It has been modified and clearly drawn in the Fig.3a.
Point 7: What kind of gases the authors tested in their study? The authors presented only simulation where refractive index of air is changed. Please, use the real refractive index dispersions of gases in your simulations. Moreover, in the introduction authors mentioned that transitional and rotational bands of molecules in the gases have absorption in the THz region. For what kind of specific gases in the 10-20 THz region such sensor would be applied?
Response 7: This article introduces the refractive index sensing of HCN gas. For gases with an absorption spectrum in the 10-20 THz band, such as N2O, gas sensing applications can be realized.
Point 8: The authors compared their simulated sensors response with presented in literature, however, the range of frequencies is rather different, what leads to detection of distinct type of gases, and sensitivity comparison become difficult. So I would like to suggest the authors to find one type of gases, which they can simulate and compared with literature.
Response 8: We discussed the real part refractive index sensing of HCN at 14THz and the imaginary part refractive index sensing of N2O near the absorption spectrum. This proves the applicability of the proposed structure. In recent theoretical terahertz sensing articles, only a few have explored sensing applications for actual gases. For example, Ref. 1, 2, and 3 all discuss the sensing sensitivity of different refractive indices, and only Ref. 7 discusses the concentration sensing of H2. It is difficult to find the relationship between the corresponding dielectric constant, concentration, and pressure of the corresponding gas, so it is hard to do practical inquiries into specific gases.
For the comparison of sensing performance, angular sensitivity and the value of Sf/f can be used. The Sθ has a good performance compared to Ref.1,2,3. But compared to Ref.7 in terms of frequency sensitivity, the value of Sf / f can only reach 1: 3.4. This is due to the higher refractive index of the photonic crystal material used in Ref.7. It is shown from line 191 to line 198.
Point 9: Would be very useful to analyse the cases with polar and non-polar gases, some references could be find here: A. Nooke et. al. Sensors and Actuators B 149 (2010) 194–198.
And a freshly published comparison of sensitivity features between BSW and SPR should be also cited:
Z. Balevicius and A. Baskys, Optical Dispersions of Bloch Surface Waves and Surface Plasmon Polaritons:Towards Advanced Biosensors, Materials 2019, 12, 3147; doi:10.3390/ma12193147
Response 9: we have added these 2 references as Ref.[13] and [20].
Point 10: Please, check grammar quite much mistakes with missing of letters.
Response 10: Sorry for this error, we have rechecked it again.

Round 2
Reviewer 1 Report
The authors have addressed the technical parts of their discussion according to the reviewers comments. The design and physical aspects is not really novel, however the discussion is solid although brief.
Author Response
Response to Reviewer Comments
We have revised "when Λ = 10μm, the effective index of refraction 1.4 < neff < 1.6, from Eq. (3). Therefore, the wave number of BSW is in the range of 1.4k0 to 1.6k0. We can get the operation frequency f for producing BSW, which lies at the range: 11.54 THz < f < 21.43 THz. It can be seen in Fig.2(a)." as shown in the contents starting from the line 109-111.
Thanks for your advise.
Reviewer 2 Report
The responces to the points 2, 7 and 8 should be discussed in the manuscript.
Author Response
Response to Reviewer Comments
Point 2: Would be useful to explain why minus sign appear in the expression (2)?
Response 2: We revised as " the negative sign “-” means the propagation direction of the BSW is opposite to that of incident k0 vector that horizontally projected. As THz wave is incident on the grating, it produces the reflected and BSW waves. As according to refs. [18], the direction of the BSW wave is opposite to the direction of the x component of the incident wave, it thus originates the negative sign as shown in Eq. (2). " as shown in the contents starting from the line 103-107.
Point 7: What kind of gases the authors tested in their study? The authors presented only simulation where refractive index of air is changed. Please, use the real refractive index dispersions of gases in your simulations. Moreover, in the introduction authors mentioned that transitional and rotational bands of molecules in the gases have absorption in the THz region. For what kind of specific gases in the 10-20 THz region such sensor would be applied?
Response 7: We revised as “According to the simulation results based on finite element method (FEM), the proposed gas sensor is sensitive to the change of refractive index real part of HCN, as indicated by the high FOM measured by frequency ~147.3 /RIU and FOM measured by the incident angle ~ 301.3°/RIU.” from line 65-68.
And added “Such as N2O, for gases with an absorption spectrum in the 10-20 THz band, gas sensing applications can be realized. ” from line 73-74.
Besides we also added the contents of HCN dielectric constant “By tuning the concentration and (or) pressure of the measured gas, the dielectric constant can be varied. Such phenomenon may be described by the equation as given as[8]:
here we select the HCN as the measured gas, note that is the background dielectric constant.is the angular frequency corresponding to the absorption peak. is the damping rate. is the absorption strength factor, which is a function of pressure P and concentration C. Given that the frequency of the incident THz wave is 14THz, the refractive index of HCN is : n0 = 1.000 +2.14×10-7i. By increasing pressure or concentration, the refractive index of the gas can be tuned, here we assume the real part of the n0 changes from 1.000 to 1.003, apparent shifts in the spectra can be found in Fig. 4(a) and Fig. 4(b), where Sθ decreases with the increase of n0, i.e. S and FOM can reach 123°/RIU and 147.3/RIU, respectively, which can come up with following formula:”as shown from line 144-154.
And we also added the contents of N2O “At present, most researches on THz gas sensor focus on sensing the change of real part of the analyte, see refs. [1-3,7]. The structure based on BSW as proposed here may also be applied for detecting the minor change of the imaginary part of the refractive index of gas. In the area of atmospheric pollution and dangerous gas detection, gases such as N2O, HCN, H2 and CO2 have obvious absorption peaks in the THz band. From Eq. (8), it can be seen that, when the operation frequency of the sensor is consistent with the absorption frequency of gas molecules, in which ω = ω0i, different concentration C and (or) pressure P will make the imaginary part of the refractive index of analyte be varied; by measuring the change of transmission in the spectra, the sensing to the imaginary part may also be achieved. Here we select the gas N2O with the absorption peak appeared in 17.664 THz (which is equivalent to the 588.726 cm-1) to detect the change of imaginary part of the N2O gas [22]. Since there is no reference that can show the relationship between and concentration C (or pressure P), in the following we consider four cases where the imaginary part is 0, 0.00002, 0.0002 and 0.002 respectively. The transmission spectrum as simulated with finite element method is shown in figure 5. It can be found that the transmittance is sensitive to the minor change of the imaginary part of the refractive index (e.g. the imaginary part changes from 0 to 0.002, the transmission changes from nearly 0 to 0.15) and the sensitivity can reach 664 (transmittance unit/ one unit of the imaginary part of the refractive index). Further optimization in terms of increasing the number of photonic crystal layers, replacing PVDF with another polymer film that shows higher refractive index may also increase the sensitivity. It turns out that our design is not only sensitive for sensing the real part of the refractive index of the gas, but also be applied as monitoring the minor change of imaginary part of the refractive index of the analyte. Consider that hazardous gases as N2O, HCN, H2 and CO2 show “finger prints” in the THz band, the proposed design may find certain applications in sensing the concentration or pressure of such gases. ” from line 167-189.
Point 8: The authors compared their simulated sensors response with presented in literature, however, the range of frequencies is rather different, what leads to detection of distinct type of gases, and sensitivity comparison become difficult. So I would like to suggest the authors to find one type of gases, which they can simulate and compared with literature.
Response 8:
We added “We discussed the real part refractive index sensing of HCN at 14THz and the imaginary part refractive index sensing of N2O near the absorption spectrum. This proves the applicability of the proposed structure. In recent theoretical terahertz sensing articles, only a few have explored sensing applications for practical gases. Performance comparison between the proposed BSW based design and other reported THz gas sensors is summarized in Table 1. Refs. 1, 2, and 3 all discuss the sensing sensitivity of different refractive indices, and only Ref. 7 discusses the concentration sensing of H2. It is difficult to find the relationship between the corresponding dielectric constant, concentration, and pressure of the corresponding gas, so it is hard to do practical inquiries into specific gases. For the comparison of sensing performance, angular sensitivity and the value of Sf / f can be used. The Sθ has a good performance compared to Refs.1,2,3. But compared to Ref.7 in terms of frequency sensitivity, the value of Sf / f can only reach 1: 3.4. It is noted that since the refractive index contrast between PC and PVDF is quite limited ~ 1, which is much smaller than the case in silicon and air (3.4 : 1) as reported in ref. [7], the normalized sensitivity of frequency Sf / f in the proposed design is small. However, the sensitivity in angle as shown in the BSW-based THz sensor is relatively high; in addition, considering the availability for sensing the minor change of both real and imaginary parts of refractive index of gases, our design may find potential applications in sensing hazardous gases in the terahertz band.” as shown from line 195-212.
